# Management of Relapsed–Refractory Multiple Myeloma in the Era of Advanced Therapies: Evidence-Based Recommendations for Routine Clinical Practice

**DOI:** 10.3390/cancers15072160

**Published:** 2023-04-05

**Authors:** Danai Dima, Fauzia Ullah, Sandra Mazzoni, Louis Williams, Beth Faiman, Austin Kurkowski, Chakra Chaulagain, Shahzad Raza, Christy Samaras, Jason Valent, Jack Khouri, Faiz Anwer

**Affiliations:** 1Department of Hematology and Medical Oncology, Cleveland Clinic Foundation, Taussig Cancer Institute, Cleveland, OH 44106, USA; 2Department of Pharmacy, Cleveland Clinic Foundation, Cleveland, OH 44195, USA; 3Department of Hematology and Medical Oncology, Cleveland Clinic Foundation, Maroone Cancer Center, Weston, FL 33331, USA

**Keywords:** relapsed/refractory multiple myeloma, immunomodulators, proteasome inhibitors, immunotherapy, targeted therapy, CAR-T cell therapy, bispecific antibodies, autologous stem cell transplant

## Abstract

**Simple Summary:**

Multiple myeloma is an incurable hematologic malignancy arising from terminally differentiated B-cells. Over the last decade, advancements in therapeutics have radically shifted the treatment landscape of MM, significantly improving the survival of patients. In this review, we will discuss the available therapeutic modalities for relapsed–refractory disease, highlighting critical factors that guide the therapy selection process.

**Abstract:**

Multiple myeloma (MM) is the second most common hematologic malignancy in adults worldwide. Over the past few years, major therapeutic advances have improved progression-free and overall survival, as well as quality of life. Despite this recent progress, MM remains incurable in the vast majority of cases. Patients eventually relapse and become refractory to multiple drug classes, making long-term management challenging. In this review, we will focus on the treatment paradigm of relapsed/refractory MM (RRMM) in the era of advanced therapies emphasizing the available novel modalities that have recently been incorporated into routine practice, such as chimeric antigen receptor T-cell therapy, bispecific antibodies, and other promising approaches. We will also discuss major factors that influence the selection of appropriate drug combinations or cellular therapies, such as relapse characteristics, and other disease and patient related parameters. Our goal is to provide insight into the currently available and experimental therapies for RRMM in an effort to guide the therapeutic decision-making process.

## 1. Introduction

Multiple myeloma (MM) is the second most common hematologic malignancy in adults. Despite the expanding therapeutic armamentarium, it unfortunately remains incurable, with a 5-year survival rate of approximately 58% [1]. Over the past few decades, the therapeutic landscape has dramatically evolved, improving the depth and duration of response that over time has translated into prolonged survival outcomes [2]. Treatment options now include combination regimens of systemic plasma-cell directed agents, targeted therapies, autologous hematopoietic cell transplant (AHCT) and, most recently, genetically modified cellular therapies and bispecific antibodies [3,4,5,6]. Given this increasing variety of available modalities, the sequenced treatment algorithm of MM has become very perplexing. In this review, we present the available therapeutic options for relapsed/refractory multiple myeloma (RRMM) currently used in routine clinical practice (Figure 1). We further discuss critical factors that should be considered prior to selection of the therapeutic schema, always in the setting of a tailored care plan based on a patient’s specific characteristics, comorbidities and needs.

## 2. Definition of Relapsed and/or Refractory Disease

Relapsed MM is defined as progressive disease after the acquisition of a response to prior therapy. Relapse can be biochemical, radiographic and/or clinical in nature. Per the International Myeloma Work Group (IMWG) criteria, relapse/progressive disease (PD) is defined as 25% increase from the lowest response value in any of the following:▪Serum monoclonal (M)-protein (absolute increase must be ≥0.5 g/dL). An M-protein increase of ≥1 g/dL indicates PD, if the lowest M-protein value is ≥5 g/dL.▪Urine M-protein (absolute increase must be ≥200 mg/24 h).▪Difference between the involved and uninvolved free light chains (FLC) (absolute increase must be >10 mg/dL). This criterion should only be used for patients who lack measurable M-protein in the serum and urine, which is defined as serum M-protein <1 g/dL and urine M-protein < 200 mg/24 h.▪Bone marrow plasma cell percentage (absolute increase must be ≥10 percent). This criterion should only be used for patients who lack measurable serum and urine M-protein levels and additionally lack measurable involved FLC levels.

In addition, PD is also defined as development of new soft tissue plasmacytomas or new bone lesions, or ≥50% increase (and at least 1 cm) in size of any existing plasmacytoma or bone lesion. Clinical relapse is defined as an increase in serum corrected calcium > 11.5 mg/dL, increase in serum creatinine ≥ 2 mg/dL (attributable only to myeloma), decrease in hemoglobin by ≥2 g/dL or development of hyperviscosity related to serum M-protein.

Refractory disease is defined as disease that never achieved even minimal response with therapy. Finally, relapsed and refractory disease is defined as disease that had at least a minor response, but then either became non-responsive while undergoing salvage therapy or progressed within 60 days of last therapy [7,8].

## 3. Available Therapeutic Modalities

### 3.1. Proteasome Inhibitors

During the past few decades, proteasome inhibitors (PIs) and immunomodulatory drugs (IMiDs) have been established as the major cornerstones in the treatment paradigm of MM. Duplet or triplet combinations of both drug classes along with steroids are frequently used for either early or late RRMM. Bortezomib (BTZ) was the first PI to be approved in 2003 followed by the next generation PI, carfilzomib (CFZ), in 2012 and ixazomib (IXZ) in 2015.

Bortezomib: BTZ can frequently cause peripheral neuropathy, which may require dose adjustments, along with other adverse effects (AEs) such as gastrointestinal disturbance and thrombocytopenia; however, it can be safely administered in renal impairment, a common phenomenon in MM [9]. The subcutaneous formulation is better tolerated from a neuropathy standpoint [10]. BTZ was historically known to have improved efficacy compared to dexamethasone (dexa) alone (APEX trial: overall response rate [ORR] 38% vs. 18%, time-to-progression [TTP] 6.2 vs. 3.5 months, both *p* < 0.001) [11]. Then, it was shown to have synergism with dexa and other agents, such as thalidomide, doxorubicin and melphalan, that have currently fallen out of favor given recent advancements. A phase I trial of BTZ with lenalidomide (LEN)-dexa yielded an ORR of 61% with overall survival (OS) of 37 months [12]. To date, BTZ is commonly used in triplets with next-generation IMiDs or monoclonal antibodies plus steroids, and sometimes cyclophosphamide (Table 1) [13].

Carfilzomib: CFZ is a next generation PI that has been mainly associated with cardiac AEs such as arrhythmias and heart failure, requiring extra caution when administered in patients with pre-existing cardiac conditions. Initial comparison of CFZ to dexa (FOCUS trial) did not demonstrate a progression-free survival (PFS) benefit [26]. However, head-to-head comparison of BTZ-dexa to CFZ-dexa showed both PFS and OS benefit for the latter arm (ENDEAVOR trial); all patients were non-refractory to BTZ [16,17,27]. The ASPIRE trial demonstrated that addition of CFZ to LEN-dexa improved PFS and OS, with OS advantage being more pronounced in patients at first relapse [14]. A subgroup analysis of the ASPIRE and ENDEAVOR trials showed that addition of CFZ improved ORR and PFS, regardless of whether patients had an early or late relapse following their most recent prior therapy [28]. A different subgroup analysis of the two trials found no association between cytogenetic risk and reaching complete response or better (≥CR) [29]. At present, CFZ is approved as a single agent for patients who have received ≥ 1 prior therapies or as combination with 1. LEN-dexa, 2. daratumumab-dexa or 3. dexa alone for patients with 1–3 prior lines of therapy [14,15]. Dosing of CFZ varies based on the drug combination and tolerance. The CHAMPION study established the once weekly dosing of CFZ when used with dexa, with maximum dose being 70 mg/m^2^ once weekly [30]. The phase III ARROW trial compared the high once-weekly (70 mg/m^2^) to the low twice-weekly dosing (27 mg/m^2^), noting that the median PFS was higher in the once-weekly arm (11.2 vs. 7.6 months, *p* = 0.0029). Grade ≥ 3 AEs were higher in the once-weekly arm; however, cardiac toxicity of grade ≥ 3 was lower in the same arm, suggesting that once-weekly dosing is safe and possibly more effective [31,32].

Ixazomib: IXZ is an orally bioavailable PI that was approved in combination with LEN-dexa in RRMM patients with ≥1 prior line of therapy, based on the results of the phase III Tourmaline-MM1 study, which showed that addition of IXZ led to a significantly longer PFS, including patients with high-risk cytogenetics. However, over time, this did not translate into a significant OS benefit [18,19]. IXZ is well tolerated and has not been linked with neurologic or cardiac AEs. A recent phase II study combined IXZ with cyclophosphamide-dexa, showing an ORR of 48% and a median PFS of 14.2 months [33]. A recently published phase III trial compared the duplets of IXZ-dexa vs. pomalidomide (POM)-dexa in CFZ and/or BTZ-exposed/intolerant, and LEN-refractory patients with ≥2 prior lines of therapy. Median PFS was similar in both groups, with the authors concluding that IXZ-dexa represents an important LEN-free, oral option for heavily pretreated, LEN-refractory, PI-exposed patients [34].

### 3.2. Immunomodulators

LEN and POM are the main IMiDs used in clinical practice at present [35,36,37]. Thalidomide is a historic first generation IMiD that has currently fallen out of favor given the high rate of accompanying neuropathy, considerably lower activity compared to next generation IMiDs and recent therapeutic advancements. Table 1 highlights the major phase II and III clinical trials of triplet combinations including a PI, an IMiD and steroids in the relapsed/refractory setting.

Lenalidomide: Early studies of LEN along with dexa (MM-009 and MM-010 trials) showed remarkable ORR (60.6% vs. 21.9%) and PFS (13.4 vs. 4.6 months) when compared to dexa and placebo [38,39,40]. Subsequent studies revealed synergism of LEN with PIs. The combination of LEN with BTZ-dexa yielded good (ORR 64%, PFS 9.5 months) results in the relapsed/refractory setting [41,42]. LEN is frequently associated with cytopenias (that may require dose adjustments) and gastrointestinal symptoms, especially diarrhea and secondary malignancies; therefore, close monitoring is necessary. Other important AEs which are universal for this drug class are volume overload and increased risk of venous thromboembolism; therefore, prophylaxis with either anticoagulation or antiplatelet agents is strongly recommended by major oncologic societies [43,44]. All IMiDs require dose adjustment in the presence of renal failure.

Pomalidomide: POM is the latest IMiD approved after the completion of the phase II MM-002 trial, that assessed the efficacy and safety of POM with or without low-dose dexa in patients who had previously received LEN and BTZ [37]. The subsequent MM-003 (NIMBUS) phase III trial demonstrated that combination of POM-dexa had superior outcomes compared to high-dose dexa (ORR: 30% vs. 9%; PFS: 4 vs. 1.9 months [*p* < 0.001]; OS: 12 vs. 8 months [*p* = 0.0234]) in heavily pretreated MM patients (median of five prior therapies), of whom the vast majority was refractory to LEN (93%) [35]. POM was then assessed in combination with newer agents such as BTZ-dexa (ORR: MM-005 phase I 75%, phase I/II 86%) [45,46,47], CFZ-dexa (ORR: phase I 50%, HOVON114 phase II 92%) [22,23,24,25], cyclophosphamide-dexa (phase II, ORR 65–86%) [48,49,50] and IXZ-dexa (phase I/II, ORR 48–52%) [51,52], demonstrating promising efficacy. The OPTIMISMM phase III trial demonstrated that the addition of POM to BTZ-dexa improved survival outcomes including for patients with high-risk cytogenetics; 70% of patients were LEN refractory [20,21] (Table 1). Most common toxicities include cytopenias and peripheral neuropathy. It is currently approved in combination with 1. dexa for patients who have received at least two prior therapies, including LEN and a PI, and have disease progression, 2. elotuzumab-dexa and 3. daratumumab-dexa, both for patients with at least two prior therapies including LEN and a PI.

### 3.3. Alkylating Agents

Cyclophosphamide (CY): CY is the oldest alkylator used in many hematologic malignancies including MM. Nowadays, it is less frequently used; however, it still remains a drug choice for heavily pretreated patients who are refractory to multiple other drug classes. CY has been studied in triplet combinations with PI, such as CFZ-dexa [53,54,55], IXZ-dexa [33,56], and with IMiDs such as POM-dexa [48,49,50,57] and LEN-prednisone [58,59]. Four treatment cycles of CY-POM-dexa at first relapse in patients who had previously received BTZ-based induction (with or without transplant), followed by LEN maintenance, showed an ORR and a very good partial response or better (≥VGPR) of 85% and 34%, respectively [49]. A quadruplet combination of CY with daratumumab-POM-dexa is also being explored and has shown some benefit [60].

Bendamustine: is a newer alkylator that has been used for heavily pretreated MM, despite not having been approved for this particular use [61]. In two phase I/II studies of patients with a median of three prior therapies, bendamustine with LEN-dexa yielded an ORR of 49% and 76%, as well as PFS of 11.8 and 6.8 months, respectively [62,63]. Its combination with POM-dexa in a more heavily pretreated population (median of seven prior therapies) yielded an ORR of 61% with 9.6 months PFS [64]. Other combinations with BTZ-dexa [65,66,67], IXZ-dexa [68] and CFZ-dexa [69] in heavily pretreated and multi-resistant patients have shown variable ORRs ranging between 40 and 70%. Combination with IXZ yielded limited response in patients previously refractory to other PIs. Combination with CFZ was particularly effective in slowing progression in patients with standard risk cytogenetics.

Melflufen: is a peptide–drug conjugate that releases alkylator payload into the tumor cells. It was initially granted accelerated approval, based on the results of the phase II HORIZON trial that evaluated melflufen-dexa in patients with a median of five prior therapies, refractory to both POM and anti-CD38 antibody, 76% of whom had triple-class-refractory disease [70]. ORR was 29% for the entire cohort and 26% for the triple-class-refractory subgroup. Median PFS and OS were 4.2 and 11.6 months, respectively. Despite these encouraging outcomes, the phase III OCEAN trial showed inferior OS of a the melflufen-based regimen compared to standard of care; therefore, melflufen was withdrawn shortly thereafter [71].

### 3.4. Targeted Therapies

Selinexor (SEL): SEL is a nuclear exportin-1 inhibitor that interrupts the nuclear-cytoplasmic trafficking, which is vital for the survival of the MM cells [72]. The phase IIb STORM trial examined the duplet of SEL-dexa in heavily pretreated patients; almost all were triple-class refractory and >50% had high-risk cytogenetics. ORR was 26% with median OS of 8.6 months [73]. These results led to the approval of the SEL-dexa combination in 2019. SEL was also evaluated with POM-dexa in the phase Ib/II STOMP study (median of three prior therapies; 49% BTZ-refractory, 88% LEN-refractory and 25% daratumumab-refractory) yielding ORR of 60%. The BOSTON phase III trial explored the addition of SEL to BTZ-dexa in patients with 1–3 prior therapies (median of one; 50% had only one prior therapy; 70% had prior exposure to BTZ and 39% to LEN), yielding an improved median PFS (14 vs. 9.5 months, *p* = 0.0066); benefit was maintained in patients with high-risk cytogenetics [74]. A phase I study evaluated the combination of SEL-CFZ in RRMM; ORR was 48% and PFS 3.7 months [75]. Notably, 95% of patients were refractory to CFZ and BTZ; ORR in CFZ-refractory patients was 62%. SEL’s AE profile is pronounced, mainly including significant gastrointestinal disturbance, fatigue and cytopenias [74].

SEL is currently approved in combination with 1. dexa for patients who have received at least four prior lines of therapy and whose disease is refractory to at least two PIs, two IMiDs and an anti-CD38 mAb and 2. BTZ-dexa in patients with at least one prior therapy. There are several ongoing trials exploring combinations of SEL with IXZ-dexa (NCT02831686), daratumumab-dexa [76] or daratumumab-BTZ-dexa [77].

Venetoclax (VEN): Bcl-2 is an anti-apoptotic protein, overexpression of which has been primarily observed in patients harboring translocation between the chromosomes 11 and 14 (t [11;14]) [78]. High levels of Bcl-2 promote plasma cell proliferation, and have been associated with poor outcomes and resistance to conventional anti-myeloma agents [79,80]. VEN is an orally bioavailable drug that selectively inhibits Bcl-2, thus disrupts the apoptotic pathway, leading to cell death. It is particularly efficacious in a subset of MM patients with the t(11;14) and high Bcl-2 gene expression [81].

VEN has been studied as monotherapy, or in combination with either dexa or BTZ-dexa for RRMM, with encouraging outcomes especially in patients with t(11;14) [82]. The BELLINI phase III trial evaluated the addition of VEN to BTZ-dexa [83]; patients were positive for t(11;14) at 10–15% and both arms had high Bcl-2 expression. The median number of prior therapies was one. The VEN arm had superior PFS (22.4 vs. 11.5 months, *p* = 0.01); however, increased mortality was noted primarily due to an increased rate of infections [83].

VEN was also studied with CFZ-dexa in a phase II trial of 49 patients (median of one prior lines of therapy) demonstrating an ORR of 80%, and ≥CR of 41%. Serious AEs occurred in 56% of the cohort, with only one death being considered as treatment-related [84]. An ongoing phase I/II study (NCT03314181) is investigating the triplet VEN-DARA-dexa with or without BTZ in 48 RRMM patients irrespective of t(11:14). Recent results from the phase I part reported grade ≥ 3 AEs in 71% and 88% of the patients in the arms with and without BTZ, respectively. The ORR was >90% for both arms, without any treatment-related deaths [85]. While VEN is not yet approved by the Food and Drug Administration (FDA), the National Comprehensive Cancer Network (NCCN) guidelines recommend VEN-dexa for only t(11;14) RRMM patients. Several ongoing trials are exploring VEN with other agents (NCT02899052, NCT03539744, NCT03567616, NCT03732703).

### 3.5. Traditional Immunotherapy

#### 3.5.1. Monoclonal Antibodies

Monoclonal antibodies (mAbs) target antigenic epitopes primarily located on the surface of plasma cells and lead to cell death via various mechanisms. Currently mAbs against the CD38 glycoprotein and signaling lymphocyte activating molecule family-7 (SLAMF7) have been developed and broadly used in routine clinical practice.

Daratumumab (DARA): DARA is the first human mAb that targets the CD38 transmembrane glycoprotein that is overexpressed on the surface of plasma cells. The binding of DARA to CD38 leads to cell death via various mechanisms including complement-dependent cytotoxicity (CDC), antibody-dependent cellular cytotoxicity (ADCC), antibody-dependent cellular phagocytosis (ADCP) and apoptosis via crosslinking [86,87]. DARA has been widely used in routine clinical practice given its high efficacy, with a relatively benign toxicity profile [88]. The initial phase II SIRIUS trial reported an ORR of 29.2% in patients with a median of five prior therapies, who were treated with single agent DARA [89]. Of these, 95% were refractory to PIs and IMiDs. The GEN503 trial combined DARA with LEN-dexa with a remarkable ORR of 81% [90].

The landmark phase III CASTOR and POLLUX trials investigated the synergism of DARA in combination with BTZ-dexa and LEN-dexa, respectively, in the relapsed setting. The addition of DARA significantly improved ORR and PFS leading to the FDA approval of these combinations (Table 2). Prolonged PFS and improved responses were sustained in patients with both high and standard cytogenetic risk; however, benefit was less pronounced for the high-risk group in the CASTOR trial [91,92]. The PFS benefit in CASTOR was seen in patients with prior exposure to BTZ, thalidomide or LEN, and in LEN-refractory patients, and was more prominent in patients with one prior line of therapy [93]. Likewise, the OS benefit was also more pronounced in patients with one prior line of therapy [94]. A recent post hoc analysis of both trials assessed whether the addition of DARA can lead to deep response based on timing of relapse in patients who had received one prior line of therapy. Results showed that DARA led to high rates of ≥CR in both early and late relapse groups [95]. In both trials, DARA increased the minimal residual disease (MRD) negativity rate, including patients with high-risk cytogenetics. Among patients with MRD-positive status, the addition of DARA significantly prolonged PFS in both studies [96,97].

DARA was then evaluated with POM-dexa and CFZ-dexa in RRMM patients, most of whom were refractory to BTZ and/or LEN, with ORR of 60% and 84%, respectively. The phase III CANDOR trial compared the triplet DARA-CFZ-dexa to CFZ-dexa alone. Addition of DARA demonstrated superior outcomes, including the subgroup of LEN-refractory patients (Table 2) [98,99,100,101]. OS data are not yet mature. Similarly, the APOLLO trial explored the addition of DARA to POM-dexa and concluded that the triplet therapy reduced the risk for disease progression or death compared to POM-dexa alone [102].

DARA has additionally been studied in various quadruplets. Two phase II trials combined DARA with POM-CFZ-dexa, and reported an ORR of 86% and 95%, respectively [103,104]. The combination of DARA with IXZ-POM-dexa is also under investigation in a phase II trial with 47% of the cohort harboring high-risk cytogenetics. Median PFS and OS were 9.5 and 39 months, respectively [105]. Another phase II trial evaluated DARA with oral CY-dexa with (arm A) or without (arm B) POM, with ORR of 88.5% in arm A and 50.8% in arm B [60].

At present, DARA is approved: 1. as monotherapy in patients with at least three prior therapies including a PI and an IMiD or refractory to both PI and IMiD, 2. in combination with LEN-dexa or BTZ-dexa in patients with at least one prior therapy, 3. in combination with CFZ-dexa in patients with 1–3 prior lines of therapy and 4. in combination with POM-dexa in patients with at least two prior lines of therapy, including LEN and a PI.

Isatuximab (ISA): ISA is a chimeric anti-CD38 mAb that works similarly to DARA, but binds to a different epitope of the CD38 molecule. It is currently approved in combination with 1. POM-dexa for the patients who had ≥2 prior therapies including LEN and a PI and 2. CFZ-dexa for patients who have received ≥1 prior therapy [106,107,108,109]. The phase III ICARIA trial assessed the addition of ISA to POM-dexa showing a PFS (11.5 vs. 6.5 months; *p* = 0.001) and OS (24.6 vs. 17.7 months, *p* = 0.028) benefit; approximately 71% of patients were refractory to both LEN and a PI [106,107]. PFS and ORR advantage was maintained in the LEN-refractory, BTZ-refractory and double-refractory subgroups [110]. Similarly, the IKEMA phase II trial assessed the addition of ISA to CFZ-dexa, with initial analysis showing a PFS benefit of the ISA arm (not reached vs. 19.2 months) [108]. OS data are not yet mature. Subgroup analysis of the IKEMA trial showed the addition of ISA improved PFS and depth or response in both early and late type of relapses [111]. Ongoing clinical trials are evaluating ISA with other regimens in the relapsed/refractory setting (NCT04126200, NCT04643002, NCT01749969, NCT03989414, NCT04083898).

Elotuzumab (ELO): ELO is a human mAb that binds to the SLAMF7 receptor on the surface of plasma cells and simultaneously binds to the CD16 receptor of natural killer (NK) cells via its Fc portion, forming a bridge between the MM and NK cells. This interaction activates NK cells to destroy the malignant MM cells via ADCC dependent or independent manner [112,113]. Initial clinical trials only showed modest activity of the ELO as a single agent. However, its combination with LEN or POM yielded significant efficacy, leading to the ELOQUENT-2 phase III trial which assessed the addition of ELO to LEN-dexa in the relapsed/refractory setting. All patients were LEN-naïve. The addition of ELO resulted in a higher ORR, improved median PFS and OS [114]. Likewise, the ELOQUENT-3 phase II trial evaluated the addition of ELO to POM-dexa in patients refractory to LEN and a PI. Addition of ELO again led to superior outcomes, including in the subgroups of patients who were double-refractory to both a PI and an IMiD, and those who were heavily pretreated. Outcomes from a combination of ELO-BTZ have been disappointing to date [115]. ELO is currently studied in combination with POM-BTZ-dexa (NCT02718833) and POM-CFZ-dexa (NCT03155100). It is approved in combination with 1. LEN-dexa after ≥1 prior therapy and 2. POM-dexa after ≥2 prior therapies including LEN and a PI.

Most frequent AEs of mAbs include fatigue, headache, nasal congestion, throat irritation, chills and fevers. Administration-related reactions are also common, usually mild (grade 1–2), and the vast majority occur with the first dose, therefore requiring premedication with steroids, acetaminophen and antihistamines. Hematologic toxicity includes anemia, neutropenia, thrombocytopenia and leukopenia. MAbs are immunosuppressive thus increase the risk for infections especially upper and lower respiratory (particularly anti-CD38 agents). Prophylaxis against herpes zoster reactivation with acyclovir or valacyclovir is recommended for all patients receiving mAbs. No dose adjustments are recommended in patients with severe renal impairment; however, available data regarding safety in patients with severe renal insufficiency or on dialysis is limited, as these patients were excluded from major trials.

#### 3.5.2. Antibody Drug Conjugates

Antibody drug conjugates (ADCs) are mAbs against a specific target on the surface of plasma cells that carry a small cytotoxic agent (payload). When it reaches its target, the ADC is internalized releasing its payload into the cytoplasm of malignant cells leading to cell death [116]. The most well studied ADC target on plasma cells is the B-cell maturation antigen (BCMA) [117]. The anti-BCMA ADC belantamab mafodotin (blenrep) is a humanized IgG1 mAb [118]. Its clinical safety and efficacy were assessed by the phase I/II DREAMM-1 and DREAMM-2 trials in a heavily pretreated population. Encouraging outcomes led to its accelerated approval as a single agent, for patients who have received at least four prior therapies including an anti-CD38 mAb, a PI and an IMiD [119,120,121,122]. A 13-month follow-up analysis of the DREAMM-2 showed an ORR of 32%; median PFS and median OS were 2.8 and 13.7 months, respectively. All patients were refractory to DARA, BTZ and LEN and had a median of 6–7 prior lines of therapy [123]. Notably, responses were poor in patients with extramedullary disease.

The major AE of blenrep is ocular toxicity/keratopathy, an off-target AE, which was noted in approximately 80% of the trial patients. For this reason, the FDA implemented the risk evaluation and mitigation strategy (REMS) which mandates patient evaluation by an ophthalmologist with each cycle of blenrep therapy. This strict requirement appeared to limit the general enthusiasm and uptake in utilization among providers. No other significant AEs were noted and the drug was well tolerated making it a good option for elderly patients. Importantly, it can be safely used in renal failure. Despite this and further encouraging results [124,125,126,127,128], in November 2022 blenrep was withdrawn from the market due to the final results of the DREAMM-3 phase III trial, which compared single agent blenrep to POM-dexamethasone, but failed to meet its primary endpoint which was PFS superiority (1-year PFS: 11.2 vs. 7 months). Additional trials are currently assessing blenrep in combination with other novel agents.

Other anti-BCMA ADCs are currently under investigation including AMG224 (NCT02561962), MEDI2228 (NCT03489525), CC-99712 (NCT04036461) and HDP-101 (NCT04879043) [129,130]. Other targets of ADCs, apart from BCMA, include the CD38, CD46 and CD74 transmembrane proteins on the surface of plasma cells. Investigational drugs are currently being tested at a preclinical and clinical level; however, none has been approved for use in humans yet [131,132,133,134].

### 3.6. Advanced Immunotherapy

#### 3.6.1. Chimeric Antigen Receptor T-Cell Therapy (CAR-T)

Chimeric antigen receptors (CARs) are synthetic transmembrane protein receptors that are designed to selectively recognize specific antigenic epitopes on the surface of target cells [135]. CARs are placed on the surface of physiologic T-cells, via a complex process requiring a vector, most commonly viral, to transfer the genes encoding the CAR constructs into the genome of T-cells [136]. CARs are then expressed on the surface of the physiologic T-cells forming the engineered CAR T-cells [137]. This ex-vivo process usually takes 1–6 weeks. CAR T-cell therapy is given as a single infusion after the administration of lymphodepleting chemotherapy to facilitate the CAR T-cell expansion. BCMA on MM cells is the first antigen to be targeted in clinical trials using CAR T-cell therapy [138].

Idecabtagene vicleucel (ide-cel) is the first CAR T-cell product officially approved for heavily pretreated MM patients, followed by ciltacabtagene autoleucel (cilta-cel), both of which target the BCMA of MM cells. The initial phase I CRB-401 study demonstrated a favorable benefit–risk profile of ide-cel [139]. The subsequent phase II KarMMa study investigated the efficacy of ide-cel in 128 RRMM patients after at least 3 previous lines of therapy (median of 6 prior lines), including an IMiD, a PI and an anti-CD38 mAb. Notably, 98%, 91% and 94% were refractory to IMiDs, PIs and anti-CD-38 mAb, respectively. Also, 89%, 84% and 26% of patients had double, triple and penta-refractory disease, respectively. ORR was 73%, with 33% achieving ≥ VGPR. MDR negativity (10^−5^) was confirmed in 26% of the entire cohort; 79% of the patients who achieved ≥ CR were MRD negative [140]. Median PFS was 8.8 months. KarMMa-3 is a phase III study that compared the efficacy of ide-cel to standard regimens in 386 RRMM patients who had received 2–4 prior therapies including an IMiD, PI and DARA (triple class exposed) and who had refractory disease to their last regimen. Of these, 66% had triple-class-refractory and 95% DARA-refractory disease [141]. ORR (71% vs. 42%, *p* < 0.001), ≥CR rate (39% vs. 5, *p* < 0.001) and median PFS (13.3 vs. 4.4 months, *p* < 0.001) were all superior for the ide-cel arm. OS data are not yet mature [141].

Recent real-world data reported ORR and ≥CR of 84% and 42%, respectively, in 159 patients treated with ide-cel; notably, 75% of the cohort did not meet the eligibility criteria for the KarMMa trial. After a median follow up of 6.1 months, median PFS and OS were 8.5 and 12.5 months, respectively. Patients with previous exposure to anti-BCMA therapy, high-risk cytogenetics, Eastern Cooperative Oncology Group performance status ≥ 2 at lymphodepletion and young age had inferior PFS. Authors concluded that outcomes of ide-cel in a real-world setting were comparable to those reported by KarMMa, confirming its safety and efficacy [142].

Bb21217 is another anti-BCMA CAR T-cell product that uses the same CAR molecule as ide-cel but adds the PI3K inhibitor, bb007, to enrich the product in memory-like T-cells. The CRB-402 is an ongoing phase I dose escalation trial evaluating bb21217 in patients who have received ≥ 3 prior regimens, including a PI and IMiD, or are double-refractory to both classes. In the expansion cohort, patients additionally required prior exposure to an anti-CD38 mAb. To date, 72 patients have been enrolled with a median of 6 prior lines of therapy, 56% of whom were triple class refractory. Reported ORR was 69% with 28% of patients achieving ≥ CR and 58% ≥ VGPR [143].

The CARTITUDE-1 phase I/II study assessed the efficacy of cilta-cel in 97 patients with ≥3 previous lines of therapy (median of 6 lines), including a PI, IMiD and anti-CD38; 84% of patients were penta-drug exposed and 88% were triple-class refractory [144]. ORR was impressive at 89%, with 67% achieving ≥CR. Median PFS was not reached after a median follow up of 12.4 months. Only 57 patients were evaluable for MRD, since the rest of the cohort did not have an identifiable clone in the baseline bone marrow sample. Of those, 93% achieved MRD negativity (10^−5^) rapidly, with median time to MRD negative status of 1 month. An updated 2-year analysis continued to show a high ORR of 97.9%, with median PFS and OS still not reached [145,146]. Notably, duration of response and survival outcomes were shorter in patients with high-risk cytogenetics. The most common AEs in both KarMMa and CARTITUDE-1 trials were cytopenias, cytokine release syndrome (CRS) and neurotoxicity.

A single institution retrospective study, that examined the impact of day 30 MRD status in 60 patients that had previously received CAR T-cell therapy, reported that, regardless of bone marrow cellularity, MRD negativity at day 30 appears to correlate with deep response and prolonged PFS [147]. Notably, CAR T-cell therapies appear to significantly improve quality of life given that patients do not require maintenance with therapy until disease progression [148].

Other autologous CAR T-cell products are currently under rigorous investigation [149]. The CAR T-ddBCMA product, utilizing a novel synthetic binding domain, called D-Domain, demonstrated an ORR of 100% (≥VGPR 88%) after a median follow up of approximately 10 months [150]. Allogenic CAR T-cell products generated from T-cells of healthy donors have also been clinically assessed with promising outcomes and acceptable AE profiles; however, they have not been approved for clinical use yet [151,152]. These might be a good option for patients requiring prompt treatment and who cannot wait for the time-consuming manufacture of autologous CAR T-cells.

#### 3.6.2. Bispecific Antibodies

Bispecific antibodies (BsAbs) are mAbs that bind both to a target on the surface of the malignant MM cells and on the surface of the effector cells, forming an immunologic bridge leading to the destruction of the tumor cells [153]. All bsAbs, that are currently clinically tested, target the CD3 molecule on the surface of T-cells (bispecific T-cell engagers, BiTEs) [154]. Similarly to CAR-T cell therapies, major AEs of BsAbs include CRS, neurotoxicity, prolonged cytopenias, immunosuppression and frequent infections. Most frequent targets of the BiTE currently tested in clinical setting, is the BCMA of MM cells and CD3 on T cells (BCMAxCD3).

Teclistamab (BCMAxCD3) is the only BiTE approved for clinical use as monotherapy after the interim analysis of the phase I/II MajesTEC-1 study. Teclistamab use in 165 patients with ≥3 prior lines of therapy (median of 5), yielded an ORR of 63%, with 39.4% achieving ≥ CR and 26.7% MRD negativity rate, after a median follow up of 14.1 months [155,156]. All patients were exposed to a PI, IMiD and anti-CD38. The majority of patients (77%) were triple-class refractory. Median duration of response was 18.4 months and median PFS was 11.3 months. The most common AEs were CRS (in 72%, mainly grade 1–2), cytopenias and infections. Teclistamab was combined with DARA in the phase Ib TRIMM-2 study; the most updated analysis showed tolerable toxicity and encouraging efficacy (ORR 78%, ≥ VGPR 73%) [157]. A recent indirect comparison of teclistamab vs. real-world treatment for triple exposed RRMM reported that teclistamab yielded improved ORR, PFS and OS [158]. MajesTEC-3 is a phase III trial currently recruiting, with the aim to assess teclistamab-DARA vs. investigator’s choice in RRMM [159].

Talquetamab targets the GPRC5D molecule instead of BCMA on the surface of MM cells (GPRC5DxCD3). The phase I MonumenTAL-1 trial assessed talquetamab monotherapy in 232 heavily pretreated patients (median of 6 prior lines of therapies); 79% had triple-class-refractory disease, 30% penta-drug-refractory disease and 87% disease refractory to the last line of therapy. After a median follow up of 11.7 months (for patients who received the 405-μg dose level) and 4.2 months (for patients who received the 800-μg dose level), ORR was 70% and 64%, respectively, with median duration of response being 10.2 months and 7.8 months, respectively [160,161,162].

Other BiTEs tested in a clinical setting include AMG420; a phase I dose escalation study of 42 patients generated an ORR of 31%; however, at maximum tolerated dose of 400 μg/d the response rate was 70% with 50% being MRD-negative CR [163]. The MagnetisMM-1 phase I trial reported that elranatamab (BCMAxCD3) yielded an ORR of 64% in heavily pretreated patients, with only grade 1–2 CRS in 67% of the trial population [164]. The MagnetisMM-3 phase II trial also reported good tolerance of elranatamab in triple-class refractory patients, however, efficacy outcomes are not yet available [165]. Similarly, the MagnetisMM-9, phase I/II study, is evaluating an alternative elranatamab dosing in an effort to further mitigate CRS with results pending. MagnetisMM-5 is a three-arm phase III randomized trial (NCT05020236) currently recruiting, with the goal to assess elranatamab monotherapy, elranatamab-DARA vs. DARA-Pd in RRMM. ABBV-38 is another BCMAxCD3 BiTE, currently studied in an ongoing phase I trial, that has demonstrated an ORR of 57% with ≥VGPR of 43% in heavily pretreated patients (median of five prior therapies) [166]. Initial reports on the efficacy of BsAbs have been impressive so far, considering the high ORR and prolonged PFS in a heavily pretreated RRMM population; long-term data including OS are eagerly awaited.

### 3.7. Salvage Transplant

Salvage autologous hematopoietic cell transplantation (sAHCT) is an option for patients who have either already undergone AHCT and have experienced long remission, or patients who deferred upfront AHCT. A phase III study (ReLApsE) compared sAHCT followed by LEN-maintenance vs. LEN-dexa, during 1–3 relapses and did not show significant benefit with sAHCT (ORR: 75% vs. 78%) [167]. However, there was a trend for improved PFS and OS in the sAHCT group [168]. Most (94%) patients had upfront AHCT. The subgroup of patients with high-risk cytogenetics experienced improved OS after sAHCT. The phase III Myeloma X study compared second sAHCT to oral CY maintenance after both arms received re-induction with BTZ-doxorubicin-dexa; the transplant arm had significant PFS (19 vs. 11 months, *p* < 0.0001) and OS (67 vs. 52 months, *p* = 0.0169) benefit [169]. A retrospective study of 975 patients who underwent second sAHCT showed that those who relapsed after ≥36 months from first AHCT had a significantly lower relapse rate after second AHCT (3-year PFS: 16% vs. 9%; *p* = 0.01) and superior OS (3-year OS, 72% vs. 61%; *p* = 0.004), compared to those relapsing within 24–35 months of first AHCT [170]. Disease status prior to second AHCT was the only prognostic factor; ≥VGPR achievement yielded superior PFS. Another retrospective study assessed 44 patients who underwent second sAHCT and received CFZ-LEN-dexa as re-induction regimen. Post re-induction, 57% achieved ≥VGPR, which increased to 77% post sAHCT. Median PFS was 23.3 months. Patients who achieved ≥VGPR post sAHCT and patients who received maintenance treatment post-sAHCT had superior PFS [171].

The efficacy of sAHCT in patients with 1–3 prior lines of therapy (86% at first progression) is currently assessed in an ongoing phase II trial. Of the 23 enrolled patients, 36% harbor high-risk cytogenetics and 86% had a prior AHCT. All patients received four cycles of DARA-CFZ-LEN-dexa before and after AHCT, followed by maintenance. Interim analysis (82% of the cohort had received AHCT, 59% had completed all study treatments) showed CR in 45%, >VGPR in 77% and >PR in 82% patients, respectively [172].

A major challenge is the scarce data regarding type and duration of re-induction prior to sAHCT for RRMM. To date, there are some encouraging data for POM-CY-dexa and POM-CFZ-dexa. A phase II trial evaluated the efficacy of POM-CY-dexa as re-induction therapy at first relapse, prior to sAHCT. Patients had previously received BTZ-LEN-dexa induction, with (arm B) or without (arm A) upfront AHCT, followed by LEN maintenance [50]. Per study design, patients in arm A who respond to re-induction would undergo sAHCT. ORR and ≥VGPR for the entire cohort were 85% and 34%, respectively after four cycles of POM-CY-dexa. From arm A, 94% of patients could proceed to sAHCT; from arm 2, seven patients ultimately proceeded to second sAHCT. POM-CFZ-dexa was also evaluated as re-induction in a phase II trial, where all participants were in their first relapse and had previously received induction with BTZ-based triplet regimen, followed by LEN maintenance. AHCT was performed in the majority of patients who had not previously received AHCT after eight cycles of POM-CFZ-dexa [50].

There are no guidelines regarding the optimal timing and candidacy for sAHCT. However, several societies recommend the use of second sAHCT only in relapsed patients whose initial remission post-AHCT lasted ≥18–24 months [173]. More prospective data are needed to better define the role of sAHCT in the era of novel therapies.

## 4. How to Approach Therapy Selection

Patient, disease and treatment-related factors should be considered when selecting therapy for RRMM [174]. Patient-related factors include age, comorbidities, frailty and performance status. Disease-related parameters refer to the nature of the disease—risk status, organ damage, duration and depth of response to prior treatments, and relapse characteristics, such as biochemical vs. clinical relapse and rapidity of relapse. Treatment-related factors refer to type of prior regimens, prior depth/duration of response, number of prior treatments, drug AEs, availability, route of administration and prior AHCT.

### 4.1. General Approach

Typically, following each relapse, the regimen selected next is given continuously until the next relapse to keep the disease under control. If patients are fit enough, three-drug regimens are always preferred over two-drug regimens, as they have consistently shown better responses and survival outcomes [175]. However, frail patients may only be able to tolerate doublet therapy. Another approach is initiation of a three-drug regimen, with de-escalation to a single agent maintenance, if good control is achieved, until disease progression. Before treatment selection, exposure and refractoriness to prior agents should also be considered. Generally, it is preferable to start a drug class that has not been used before if possible (some patients are heavily pretreated and exposed to all drug classes). If a class is reused, then a different agent should be chosen. Additionally, some patients are exposed but not refractory to certain drugs; these could be repeated at later relapses, with perhaps a different combination [176,177].

### 4.2. High Risk—Aggressive Disease

Parameters indicating that a relapse is aggressive include the timing of relapse (early vs. late), the type of relapse (clinical vs. biochemical) [178], and/or the presence of plasma cell leukemia [179], extramedullary disease [180,181], renal insufficiency or new bony lesions. Typically relapse that occurs early (regardless of cytogenetic risk), especially within 12 months from prior AHCT or 18 months from initial therapy, and clinical relapse with any of the above presentations, is suggestive of aggressive disease associated with poor outcomes [182,183]. Shorter duration of response to a prior line of therapy also indicates aggressive disease.

Available scoring systems that risk stratify MM include the International Staging System (ISS) which uses the serum β-2 microglobulin and albumin at diagnosis [184]. A revised version, R-ISS, additionally incorporates high-risk chromosomal abnormalities (del[17p], t[4;14] and t[4;16]) and serum lactate dehydrogenase [185]. A modified version, R2-ISS, also includes the acquisition of extra copies of the chromosome 1q as a high-risk factor [186,187]. Although these scoring tools are usually applied at time of diagnosis, high-risk features may arise later in the disease course. Patients with stage III R-ISS disease at diagnosis have poor survival outcomes when they relapse [188]. Acquisition of 17p deletion at relapse is also a poor marker [189]. At present, no regimen is specifically recommended for high-risk disease; however, common clinical practice is that the most effective therapies should be implemented early. Three drug regimens should be used in patients with high-risk disease, preferably containing an anti-CD38, if patient is not refractory. Other agents preferred for high-risk relapsed disease are at least one next generation PI and/or IMiD (POM and/or CFZ), as they are considered more potent than their predecessors. Four-drug combinations may be considered for young fit patients with high-risk relapsed disease such as DARA-POM-CFZ-dexa or DARA-POM-CY-dexa or ELO-POM-BTZ-dexa [103,104,190].

CAR T-cell therapy is under investigation for early aggressive relapse in young individuals. The phase II CARTITUDE-2 trial assessed the efficacy and safety of cilta-cel in 19 patients with 1 prior line of therapy with high-risk disease and early relapse defined as <12 months after AHCT or from initial therapy if not transplanted upfront. ORR was 100%; 90% achieved ≥CR, with 12-month PFS rate being 90% [191]. The same trial also assessed the efficacy of cilta-cel in 20 patients after 1–3 prior lines of therapy; all were LEN-refractory. Again, ORR was 95%, with 85% achieving ≥ CR [192]. Similarly, the phase II KarMMa-2 study is evaluating ide-cel in high-risk MM defined as early relapse (<18 months) after frontline therapy (cohort A, B) or inadequate response after frontline AHCT (cohort C). In cohort A that included 37 patients with early relapse after AHCT, ide-cel yielded ORR of 83.8% and CR of 45.9%. Most patients were refractory to an IMiD (96.5%) or PI (89.2%), and 86.5% were double-refractory. Median PFS was 11.4 months [193]. In cohort C ide-cel yielded ORR of 87.1% with CR rate of 74.2% in 31 patients. The 12- and 24-month PFS rates were 90.1% and 83.1%, respectively [194]. Given these impressive outcomes it is likely that CAR T-cell therapies may be approved for earlier relapses in patients with high-risk features.

For very aggressive relapse accompanied by high disease burden and extramedullary sites, cytotoxic containing drug regimens such as VTD-PACE (dexamethasone, cyclophosphamide, etoposide and cisplatin) [195,196,197], or DCEP (dexamethasone, cyclophosphamide, etoposide and cisplatin) may be given for a limited number of cycles to quickly control the disease [198]. For patients who are candidates and have stored collected stem cells, but differed upfront transplant, if good control is achieved with intensifying plasma-cell directed therapy, sAHCT is a reasonable option.

Extramedullary disease (EMD) is a marker of aggressive MM and can be very heterogenous with regard to location ranging from soft tissue, bone and organ plasmacytomas (PC) to plasma cell leukemia with dismissal prognosis [181,199]. Treatment of EMD is challenging as these patients have been excluded from major clinical trials. Again, no specific regimen is recommended; general rules for high-risk disease should be followed here, including utilization of next generation PIs and/or IMiDs [200,201] combined with an anti-CD38 agent. More recent data have demonstrated the efficacy of anti-BCMA CAR T-cell therapy against EMD. Local radiation should also be considered for palliative purposes and symptom control including pain.

### 4.3. Comorbidities and Major Organ Dysfunction

In addition to age, approach to therapy is largely impacted by frailty and organ function that predict tolerance to different therapies. Frailty assessment includes medical comorbidities, cognitive health and fitness level. The choice of regimen should also be guided by the toxicity profile.

Cardiac disease is an important comorbidity to consider in patients receiving CFZ, which is particularly cardiotoxic in older individuals and those with pre-existing cardiovascular disorders including heart failure and arrhythmias [202,203]. Increasing infusion time or reducing dosage per treatment time can be helpful. CFZ should be stopped if ≥3 grade cardiac AEs occur. Treatment may be re-started based on the patient’s recovery and risk–benefit assessment. BTZ, IXZ or anti-CD38 mAbs have not been linked to cardiac AEs. POM and LEN can cause fluid overload and, thus, should be used with caution in patients with pre-existing heart failure.

Renal impairment is another common comorbidity of patients with MM. BTZ and CFZ do not require any dose modifications, in contrast to IXZ that requires dose adjustment in patients with creatinine clearance < 30 mL/min or patients on dialysis. Similarly, mAbs do not require dose reduction and appear to be safe even in severe renal failure and in patients on dialysis; however, data are limited. LEN needs to be dose adjusted for renal failure [204,205,206]; POM can be used in renal impairment though dose adjustments may be necessary for patients on dialysis [207,208]. For patients requiring dialysis, regardless of the dose adjustment or route of administration, plasma-cell directed agents should be given after dialysis. A recent real-world multicenter experience of ide-cel in 28 patients with renal impairment reported comparable efficacy and safety to patients without renal dysfunction [209,210]. This is critical given that major CAR-T trials excluded patients with renal insufficiency.

Anti-CD38-containing regimens with ISA or DARA have been shown to be effective in patients with renal failure based on the results of subgroup analysis of the ICARIA and IKEMA trials [211,212], as well as real-world data [213,214,215]. The DARE trial assessed DARA-dexa in 38 patients with severe renal dysfunction, 50% of whom were on dialysis [216]. No unexpected toxicities were reported, with 17% of the cohort achieving renal response. These combinations may not only yield deep hematologic responses in patients with renal failure but also have the potential to reverse renal failure [217,218]. However, more data are needed to better understand the role of anti-CD38-agents in improving renal function.

### 4.4. Optimal Timing of Therapy Initiation

Type of relapse can vary among individuals, ranging from aggressive clinical presentation with new lytic lesions, plasmacytomas and low counts, to a slow indolent rise of the monoclonal protein or free light chains without any evidence of clinical deterioration. Given this heterogeneity, there are no standard rules with regards to when therapy should be initiated for relapsed disease and this decision should be tailored to each patient. For patients with high-risk features, treatment should be started immediately. The presence of a more indolent relapse without any high-risk characteristics can either be monitored closely with intervention at a later time, if and when evidence of deteriorating biochemical or clinical disease is noted, or treated early as high-risk disease, perhaps with a less intense regimen, such as a two-drug regimen [178]. There are no prospective data comparing early vs. late intervention.

### 4.5. Previous Refractoriness and Timing of Relapse

#### 4.5.1. First Relapse

After frontline therapy, regardless of AHCT, patients are usually placed on one- or two-drug maintenance therapy until disease progression. To select therapy for first relapse, it is important to consider the type of maintenance therapy. Most patients are treated with LEN maintenance and are considered LEN-refractory at first relapse. However, less frequently patients may be on BTZ or BTZ-LEN maintenance. Other significant factors to be considered include patient and disease-related characteristics, as mentioned above.

LEN-refractory patients: A highly effective option for LEN-refractory patients is POM-based regimens. Initial results of the NIMBUS trial demonstrated that POM-dexa had ORR, PFS and OS benefit in LEN-refractory patients compared to high-dose dexa alone [35]. After the introduction of anti-CD38 mAbs into clinical practice, it became evident that POM-dexa combined with either DARA or ISA yielded significant efficacy for LEN-refractory disease. In detail, the combinations of DARA-POM-dexa and ISA-POM-dexa led to improved ORR and PFS in the LEN-refractory subgroups of the APOLLO and ICARIA trials, respectively [102,106,107]. Notably, in both trials, LEN-refractory patients exceeded 80%. Survival benefit was additionally seen in the BTZ-refractory and double-refractory patients. ELO-POM-dexa is another effective regimen given that the ELOQUENT-3 trial showed a PFS and OS benefit in a population where approximately 90% were LEN-refractory. POM combined with PI is another efficacious strategy. BTZ-POM-dexa improved PFS and OS of LEN-refractory patients at their first relapse in the OPTIMISMM trial [21]. Likewise, POM-CFZ-dexa yielded a high ORR of 92% at first relapse in patients refractory to both LEN and BTZ in the EMN011/HOVON114 trial [25].

POM-free regimens that are effective in LEN-refractory MM include combinations of a PI with anti-CD38 mAbs. The CASTOR (DARA-BTZ-dexa) [91], CANDOR (DARA-CFZ-dexa) [98,100] and IKEMA (ISA-CFZ-dexa) [108] phase III trials have shown clinical efficacy, with ORR and PFS advantage in the LEN-refractory state; however, subgroups of LEN-refractory patients in these studies were relatively small and statistical significance was not always reached. Other less popular combinations include CFZ-CY-dexa which has shown a PFS benefit in LEN-refractory patients with a median of one prior therapy compared to CFZ-dexa alone (26.2 vs. 7.7 months, *p* = 0.01) [53].

Trials that have assessed the addition of a third agent (PI or mAb) to LEN-dexa doublet, such as the ASPIRE (CFZ-LEN-dexa), Tourmaline-MM1 (IXZ-LEN-dexa), POLLUX (DARA-LEN-dexa) and ELOQUENT-2 (ELO-LEN-dexa), have excluded LEN-refractory patients, so it is unclear whether these combinations are active in LEN-refractory individuals. Despite these recent advances, optimal management of LEN-refractory patients still remains an unmet need.

BTZ-refractory patients: For patients who progress on BTZ maintenance, switching drug class to a combination of IMiD plus mAb (either anti-CD38 or anti-SLAM7) is a highly effective option. IMiD-mAb-based regimens include DARA-POM-dexa (APOLLO), ISA-POM-dexa (ICARIA) and ELO-POM-dexa (ELOQUENT-3, not FDA approved yet for first relapse) which have demonstrated significant benefit in the BTZ or PI-refractory subgroups. Other alternatives include DARA-LEN-dexa (POLLUX), ELO-LEN-dexa (ELOQUENT-2), or the alkylator-containing regimens POM-CY-dexa and LEN-CY-dexa.

A different strategy for BTZ-refractory MM is the utilization of CFZ-based regimens, such as CFZ-DARA-dexa (CANDOR) [98,100], CFZ-ISA-dexa (IKEMA) [108], CFZ-LEN-dexa (ASPIRE) [14] or CFZ-CY-dexa. Notably, CFZ-LEN-dexa in the ASPIRE trial improved OS in BTZ-exposed patients at first relapse. It is unclear whether IXZ-based regimens have a role in BTZ-resistant MM. The combination of IXZ-LEN-dexa (Tourmaline MM1) showed PFS but not OS benefit in patients with 1–2 prior lines of therapy, 69% of whom were previously exposed to BTZ.

LEN-BTZ-refractory patients: As previously mentioned, for patients progressing on LEN-BTZ maintenance, the combination of POM-CFZ-dexa is a highly effective strategy at first relapse, given the results of the EMN011/HOVON114 trial showing high response: ORR 92%, 100% double-refractory patients at first relapse [25]. Other regimens showing benefit in double-refractory patients include DARA-POM-dexa, ISA-POM-dexa and ELO-POM-dexa, as well as regimens containing CY, mainly POM-CY-dexa.

For patients whose upfront therapy did not include AHCT consolidation or who underwent upfront AHCT and had a prolonged remission, then salvage AHCT should be strongly considered if institutional eligibility criteria are met. Registration to clinical trials exploring new investigational modalities or comparing known regimens/strategies at first relapse should be strongly considered. Our center’s generalized approach regarding treatment selection at first relapse is highlighted in Figure 2.

#### 4.5.2. Later Relapse

Management of patients who have received ≥2 prior lines of therapy can be very challenging as, at that point, exposure and maybe refractoriness have occurred to all drug classes. Therefore, upon later relapses, agents that have not been trialed can be utilized, such as a drug from a class that has not been introduced previously or a new agent from a class that has been tried in the past. There is evidence to support that re-treatment with the same agents at later stages of the disease may have some effect. Efficacy is more pronounced the longer the interval from previous treatment exposure; however, data are limited.

DARA-refractory patients: At present, many patients receive anti-CD38 mAb as second line of therapy; however, there is an increasing tendency to use these agents upfront resulting in anti-CD38 refractoriness at ≥first relapse. Real-world data from 275 anti-CD38 refractory patients (93% DARA-refractory; 7% ISA-refractory; 54% and 25% with triple/quad- and penta-refractory disease) reported poor prognosis with a median OS of 8.6 months [219]. ORR of next line of therapy was 31% with median PFS and OS of 3.4 and 9.3 months, respectively. Best PFS outcome was reported with combinations of CFZ-alkylator and DARA-IMiD (median PFS of 5.7 and 4.5 months, respectively) [219].

Further retrospective analyses have reported possible benefit with DARA-POM-dexa combination in patients refractory to DARA or POM or both; however, sample sizes were small [220,221,222]. A study of 43 evaluable patients reported that re-treatment with DARA-based therapy in patients who were DARA-refractory demonstrated some efficacy with ORR of 49% [223]. ELO-IMiD [224] or SEL-based regimens [73] may also be effective in DARA-refractory patients but again data are limited. On the other hand, ISA monotherapy did not show any objective responses in DARA-refractory patients with 53% achieving stable disease in a phase II trial [225]. One powerful option in this setting is CAR T-cell therapies or teclistamab, but both are approved for patients who have had at least four prior lines of therapy. It is clear from the available evidence that DARA-refractory disease is challenging to manage.

Heavily pretreated patients: In heavily pretreated patients, novel targeted therapies can be implemented, including SEL and VEN. SEL-dexa was effective in 26% of patients in the STORM study that included triple-class refractory (at least one PI, IMiD and anti-CD38) patients. Penta-refractory patients had an ORR of 25.3% with PFS of 3.8 months. SEL-BTZ-dexa can also be considered; however, this combination has been tested in a less heavily pretreated group of patients. VEN combined with BTZ-dexa can also be used in RRMM, particularly for patients harboring the t(11:14) or high BCL2 gene expression given the PFS benefit noted in the BELLINI trial [83]. ELO combined with POM-dexa is another alternative for heavily pretreated MM, yielding an ORR of 53% in patients with a median of three prior lines of therapy (all refractory to LEN and a PI) of the ELOQUENT-3 trial; however, none of these patients were refractory to POM or DARA [115].

Alkylating agents are another class that can be trialed if running out of options. Bendamustine combined with next generation IMiDs and PIs for heavily pretreated patients (median of prior therapies between 4–7) yielded ORR ranging from 41–61% and PFS 5.2–11.6 months [64,68,69]. Bendamustine in combination with LEN-dexa and BTZ-dexa, in a less heavily pretreated population (2–3 median number of prior therapies) led to an ORR between 49 and 76% [62,63,65,66,67]. CY combinations with CFZ-dexa [53,54,55], IXZ-dexa [33,56] or POM-dexa [48,49], can also be used for patients refractory to multiple other drug classes. The quadruplet combination of CY plus DARA-POM-dexa is another available option [60]. More intense regimens such as VTD-PACE [195,196,197] or DCEP [196,198] can be used in aggressive relapse to quickly achieve cytoreduction. Melflufen-dexa is another alternative that can work somewhat well in cases of EMD relapse.

Anti-BCMA novel therapies including CAR T-cell products and teclistamab have shown high efficacy in heavily pretreated patients. CAR T-cell infusion with ide-cel or cilta-cel is a one-time treatment with high rates of deep and durable responses without the need for subsequent continuous therapy. CAR T-cell therapy should preferably not be given to patients with rapidly advancing or high burden disease, as it does not offer the greatest benefit in this setting. Instead, it should be utilized in patients with slow/indolent biochemical progression or stable disease, as in these scenarios it can lead to deep and durable remissions. An important limitation of CAR T-cell therapy is the time required for manufacture. For patients requiring urgent control of their disease and who cannot wait for CAR T-cells to be manufactured, teclistamab may be used instead. Teclistamab is administered intravenously once a week after initial loading and, similarly to CAR T-cell therapy, is approved for use in heavily pretreated patients. It is unclear whether prior exposure to or progression on one anti-BCMA therapy may affect the outcome of a subsequent yet different anti-BCMA therapy, with recent real-world data reporting lower responses in this context [226,227]. Enrollment in clinical trials of novel agents should always be encouraged based on availability and candidacy. Our institution’s approach regarding treatment selection for second or later relapse is described in Figure 3.

#### 4.5.3. Relapse after CAR T-Cell Therapy

Despite the remarkable efficacy of the CAR T-cell therapy, with ORR ranging between 73 and 97%, most patients ultimately relapse. At present, there are no prospective data with regards to optimal rescue therapy selection after CAR T failure; the literature is limited to only small retrospective case series. In routine clinical practice, CAR T-cell therapy is only used in very heavily pretreated patients that have been exposed to almost all drug classes, making selection of subsequent therapy very challenging. Retreatment with other anti-BCMA agents such as blenrep has shown limited efficacy in one case series [228]. Treatment with bispecific antibodies yielded durable responses in another series; however, target of bispecific antibodies was not described and patient sample size was limited [229,230]. A recent study that explored outcomes of salvage therapies post-CAR T relapse, reported that subsequent treatment with BCMA-directed CAR T-cell therapy resulted in an ORR of 75%. Other salvage therapies such anti-BCMA bispecific antibodies, anti-CD38 mAbs and alkylator-based regimens yielded ORRs of 60%, 52.6% and 46.3%, respectively, with a median duration of therapy ranging from 1.5 to 3 months [231]. SEL or melflufen-based regimens have also shown some efficacy [232].

In our institution, SEL-CFZ-dexa combination or teclistamab are preferable options for progression after CAR T-cell therapy. In addition, enrollment of patients into clinical trials assessing non-FDA approved agents such as cereblon E3 ligase modulators (CELMoDs), magrolimab (anti-CD47 mAb), BMF219 (menin inhibitor) or other bispecific antibodies or CAR T-cell products (including allogeneic CAR T-cells) is strongly recommended. Management of these patients is a critical unmet need emphasizing the importance of developing new therapeutic strategies.

## 5. Supportive Care

### 5.1. Venous Thromboembolism Prophylaxis

The risk for venous thromboembolism (VTE) is significantly increased in patients with MM. There are several scoring systems assessing the risk of VTE in MM patients; however, the exact method to identify those with the highest risk for VTE who may benefit from aggressive pharmacologic thromboprophylaxis is still an area of debate. The most commonly utilized tools are the IMPEDE and SAVED scores which are included in the NCCN guidelines [43,44,233]. These scores incorporate patient, disease and treatment characteristics, and recommend initiation of thromboprophylaxis with either antiplatelet or anticoagulation agents based on the total score of each individual. Patients starting on IMiDs are a subpopulation of MM with particularly elevated risk for VTE; therefore, thromboprophylaxis is almost always recommended in this scenario, unless contraindicated for other reasons.

### 5.2. Infection Prophylaxis

Patients with active MM are at high risk for infections. This elevated risk has been linked to disease pathogenesis including immunoparesis, cytopenias and hypogammaglobulinemia, and treatment impact on the immune system. Recent guidelines published by the IMWG recommend prophylaxis with acyclovir (or valacyclovir) and levofloxacin in patients receiving plasma-cell directed therapies including PIs, IMiDs, mAbs, alkylators, SEL and steroids [234,235]. For those receiving prolonged courses of steroids, prophylactic trimethoprim-sulfamethoxazole is also necessary to avoid *P. Jirovecii* infection. Intravenous immunoglobulin should also be used in patients with immunoglobulin G (IgG) concentration < 400 mg/dL and recurrent infections, or those with exposure to varicella, herpes zoster and hepatitis A [235]. For patients with prolonged neutropenia, prophylaxis with granulocyte colony-stimulating factor should be strongly considered. Re-vaccinations are recommended after completion of AHCT and CAR T-cell therapy [236].

### 5.3. Management of MM-Related Bone Disease

The most recent IMWG guidelines recommend initiation of zoledronic acid (ZA) as the preferred bone-targeted agent for MM patients at clinical or biochemical relapse regardless of the presence of MM-related bone disease on imaging [237,238]. It is given monthly for at least 12 months in patients achieving ≥ VGPR. At that time, the frequency can be decreased to every 3–6 months and/or discontinued. While on ZA it is important to monitor kidney function and electrolytes.

Denosumab is an alternative option, only recommended in the presence of MM-related bone disease, and should be considered for RRMM patients with renal dysfunction [239,240]. Denosumab is given monthly and, due to its rebound effect, it is uncertain when to discontinue. If the physician–patient decision is to stop, a one-time dose of zoledronic acid should be given. Instead of completely stopping, continuing denosumab every six months should be strongly considered. If both aforementioned agents are unavailable or contraindicated, a second line option is pamidronic acid [241,242].

Due to elevated risk of osteonecrosis of the jaw, dental hygiene is important for patients on both ZA and denosumab. All patients on bone strengtheners should be on calcium and vitamin D supplements.

## 6. Conclusions

The therapeutic paradigm of RRMM has radically shifted with the introduction of novel therapeutic modalities into clinical practice, yielding not only improved survival outcomes but also improved quality of life. Increased use of CAR T-cell therapies and BiTEs in real-world practice is expected to continue enhancing outcomes of patients who were previously heavily pretreated and had a poor prognosis. In addition, implementation of these strategies earlier in the disease course is eagerly awaited. A major limitation for broader utilization of these advancements is their restricted access and application in large academic centers only, given the high complexity in the manufacturing and administration process, as well as the need for very close monitoring/follow up to identity potentially serious AEs related to these therapies in a timely manner. Financial burdens to patients are also significant and should not be overlooked.

Despite the progress made, unfortunately relapse continues to occur in the vast majority of patients, emphasizing the need for further innovations to combat this incurable disease. In addition, a subset of patients develop highly aggressive and resistant disease, unresponsive to all available therapeutics. Unraveling the genetic and epigenetic basis of the molecular evolution of MM is crucial as this may allow a more accurate risk stratification, prognostication, and prediction of therapy outcome. Most importantly, it could lead to the discovery of new strategies for prevention and potential cure of MM.

## Figures and Tables

**Figure 1 cancers-15-02160-f001:**
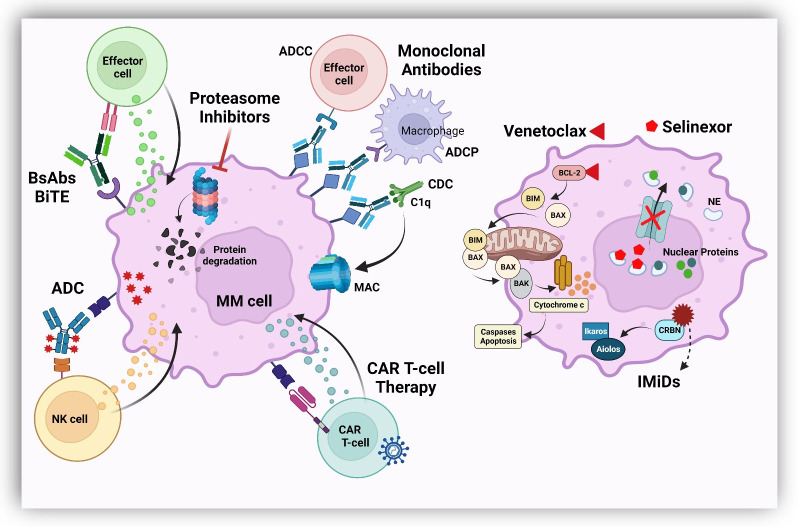
Mechanisms of action of available drug classes used in relapsed/refractory MM. Abbreviations: BsAbs, bispecific antibodies; BiTE, bispecific T-cell engager; ADC, antibody drug conjugate; IMiD, immunomodulatory drug; ADCC, antibody dependent cell cytotoxicity; ADCP, antibody dependent cellular phagocytosis; CDC, complement dependent cytotoxicity; MAC, membrane attack complex; CAR, chimeric antigen receptor; BCL-2, B-cell leukemia/lymphoma 2 protein; CRBN, cereblon; NE, nuclear export.

**Figure 2 cancers-15-02160-f002:**
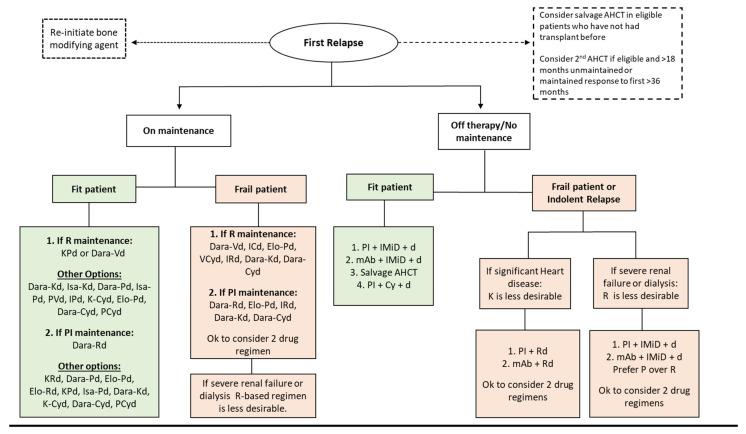
Management of MM at first relapse. Abbreviations: Cy, cyclophosphamide; Dara, daratumumab; d, dexamethasone; Elo, elotuzumab; Isa, isatuximab; V, bortezomib; K, carfilzomib; I, ixazomib; R, lenalidomide; P, pomalidomide; AHCT, autologous hematopoietic cell transplant.

**Figure 3 cancers-15-02160-f003:**
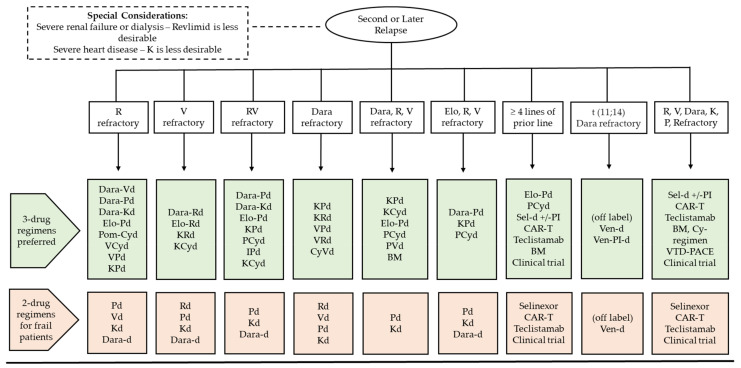
Management of MM at second or later relapse. Abbreviations: Cy, cyclophosphamide; Dara, daratumumab; d, dexamethasone; Elo, elotuzumab; Isa, isatuximab; V, bortezomib; K, carfil zomib; I, ixazomib; R, lenalidomide; P, pomalidomide; BM, bendamustine; Sel, selinexor; Ven, venetoclax.

**Table 1 cancers-15-02160-t001:** Pivotal phase II and III clinical trials of modern duplet and triplet regimens including PIs, IMiDs and/or steroids in RRMM.

Name (NCT Number)	Phase	Regimen	N	Study Population	Outcomes
ASPIRE NCT01080391	III	KRd vs. Rd	792	• Median 2 prior Tx • Exposed to V: 67% • Refractory to V: 0%	ORR: 87.1% vs. 66.7% ≥VGPR at 38% vs. 31% mPFS: 26.3 vs. 17.6 mo, *p* < 0.0001 AEs were more common in the K arm mOS: 48.3 vs. 40.4 mo, *p* = 0.0045 AEs: HF in 6.4% of the K arm [14,15] Subgroup Analysis of OS: • 1st relapse: 11.4 mo longer for K arm • 1st relapse with prior V-exposure: 12 mo longer for K arm • 1st relapse with prior AHCT: 18.6 mo Longer for K arm • ≥2 prior therapies: 6.5 mo longer for K arm
ENDEAVOR NCT01568866	III	Kd vs. Vd	929	• Exposed to V: 54% • Exposed to IMiD: ~75%	ORR: 77% vs. 63% mPFS 18.7 vs. 9.4 mo, *p* < 0.0001 [16,17] mOS: 47.8 vs. 38.8 mo, *p* = 0.0017
Tourmaline-MM1 NCT01564537	III	IRd vs. Rd	722	• Exposed to V: 70% • Refractory to R or PI: 0%	ORR: 78% vs. 72% (*p* = 0.04) ≥VGPR: 48% vs. 39% [18] mPFS: 20.6 vs. 14.7 mo, *p* = 0.01 mOS: 53.6 vs. 51.6 mo, *p* = 0.495 [19]
OPTIMISMM NCT01734928	III	PVd vs. Vd	559	• Median 2 prior Tx • Prior AHCT: 57% • Exposed to R: 100% • Refractory to R: 70%	mPFS: 11.2 vs. 7.10 mo, *p* < 0.0001 [20] mPFS of R-ref pts: 9.5 vs. 5.6 mo, *p* = 0.0008 mPFS at 1st relapse: 20.7 vs. 11.6 mo, *p* = 0.0027 [21] mOS: not available
EMN011/ HOVON114 EudraCT 2013-003265-34	II	PKd	112	• 1st relapse or primary refractory disease: 100% • Refractory to V + R: 100%	ORR: 92%, ≥VGPR 75% mPFS: 26 mo, mOS was 67 mo, 42% underwent first sAHCT, indicating that this regimen can be used as re-induction prior to delayed sAHCT [22,23,24,25]

Abbreviations: VRd, bortezomib–lenalidomide–dexamethasone; Rd, lenalidomide–dexamethasone; KRd, carfilzomib–lenalidomide–dexamethasone; IRd, ixazomib–lenalidomide–dexamethasone; PVd, pomalidomide–bortezomib–dexamethasone; Vd, bortezomib–dexamethasone; PKd, pomalidomide–carfilzomib–dexamethasone; Tx, therapies; ORR, overall response rate; mPFS, median progression-free survival; mOS, median overall survival; AEs, adverse events; HF, heart failure; sAHCT, salvage autologous hematopoietic cell transplant.

**Table 2 cancers-15-02160-t002:** Summary of major trials combining monoclonal antibodies with IMiD or PI and steroids in RRMM.

MAb	Trial	Phase	N	Regimen	Study Population	m-Prior Tx	Refractoriness	Outcomes
DARA	SIRUS	II	106	Dara single agent	• ≥3 prior Tx inc a PI and an IMiD or refractory to both PI + IMiD • Excluded pts with anti-CD38 exposure	5	• P: 63% • K: 48% • V & R: 82% • Triple: 66%	ORR: 29.2% CR: 2.8%, VGPR: 9.4%, PR: 18% m-duration of response: 7.4 months mPFS: 3.7 months 12-month OS: 64.8% mOS: 17.5 months
	CASTOR	III	498	Dara-Vd vs. Vd	• ≥1 prior Tx • Prior AHCT 61.2% • Excluded pts refractory to PI	2	• IMiD: ~33%	ORR: 83.8% vs. 63.2%, *p* < 0.0001 ≥CR: 28.8% vs. 9.8%, *p* < 0.0001 ≥VGPR: 62.1% vs. 29.1%, *p* < 0.0001 mPFS: 16.7 vs. 7.1 months, *p* < 0.0001 mOS: 49.6 vs. 38.5 months, *p* = 0.0075 MRD (−) rate: 14% vs. 2%, *p* < 0.0001
	POLLUX	III	569	Dara-Rd vs. Rd	• ≥1 prior therapies • Excluded pts refractory to R	1	• PI: ~18% • IMiD: ~5.5%	ORR: 92.9% vs. 76.4%, *p* < 0.0001 ≥CR: 56.6% vs. 23.2%, *p* < 0.0001 ≥VGPR: 80.4% vs. 49.3%, *p* < 0.0001 mPFS: 44.5 vs. 17.5 months, *p* < 0.0001 MRD (−) rate: 30.4% vs. 5.3%, *p* < 0.0001
	CANDOR	III	446	Dara-Kd vs. Kd	• 1–3 prior Tx	2	• R: 33% • V: 29%	ORR: 84% vs. 75%, *p* = 0.0080 ≥CR: 33% vs. 13%, ≥VGPR: 69% vs. 47% mPFS: 28.6 vs. 15.2 months, *p* < 0.0001 12-month MRD (−) rate: 18% vs. 4%, *p* < 0.0001
	APOLLO	III	304	Dara-Pd vs. Pd	• ≥2 prior Tx inc R and PI • Excluded pts refractory to P, or exposed to anti-CD38	2	• R: 80% • PI: 48% • PI & R: 42%	ORR: 69% vs. 46%, *p* < 0.0001 ≥CR: 25 vs. 4%, *p* < 0.0001 ≥VGPR: 51% vs. 20% <0.0001 mPFS: 12.4 vs. 6.9 months, *p* = 0.0018 MRD (−) rate: 9% vs. 2%, *p* = 0.010
ISA	ICARIA	III	307	Isa-Pd vs. Pd	• ≥2 prior Tx inc R and a PI • Excluded pts refractory to anti-CD38	3	• R: 92.5% • PI: ~75% • PI & R: ~71%	ORR: 60% vs. 35%, *p* < 0.0001 ≥VGPR: 32% vs. 9%, *p* < 0.0001 mPFS: 11.5 vs. 6.5 months, *p* = 0.001 mOS: 24.6 vs. 17.7 months, *p* = 0.028 MRD (−) rate: 7% vs. 0%
	IKEMA	II	302	Isa-Kd vs. Kd	• 1–3 prior Tx • Excluded pts previously esposed to K and pts refractory to anti-CD38	2	• R: ~32% • PI: ~33%	CR: 40% vs. 28% ≥VGPR: 73% vs. 56%, *p* = 0.0011 mPFS: 19.1 months vs. NR, *p* = 0.0007 MRD (−) rate: 30% vs. 13% *p* = 0.0004
ELO	ELOQUE NT-2	III	321	Elo-Rd vs. Rd	• 1–3 prior Tx	2	Most recent line of Tx: • V: 22% • T: 10%	ORR: 79%, vs. 66%, *p* < 0.001 ≥VGPR: 33% vs. 28% mPFS: 19.4 vs. 14.9 months, *p* < 0.001 mOS: 48.3 vs. 39.6 months, *p* = 0.0408
	ELOQUE NT-3	II	117	Elo-Pd vs. Pd	• ≥2 prior Tx inc R and a PI • Excluded pts refractory to P	3	• R: ~87% • PI: ~80% • PI & R: ~70%	ORR: 53% vs. 26% mPFS: 10.3 vs. 4.7 months, *p* = 0.008 mOS: 29.8 mo vs. 17.4 mo, *p* = 0.0217

Abbreviations: R, lenalidomide; V, bortezomib; P, pomalidomide; K, carfilzomib; d, dexamethasone; Isa, isatuximab; Dara, daratumumab; Elo, elotuzumab; PI, proteasome inhibitor; IMiD, immunomodulator; m-prior Tx, median number of prior therapies; Tx, therapies; pts, patients; inc, including; ORR, overall response rate; CR, complete response; VGPR, very good partial response; mPFS, median progression-free survival; mOS, median overall survival; MRD, minimal residual disease.

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
