# Peer review of "Management of Relapsed–Refractory Multiple Myeloma in the Era of Advanced Therapies: Evidence-Based Recommendations for Routine Clinical Practice"

_cancers, 2023, doi:10.3390/cancers15072160_

Round 1

Reviewer 1 Report

It is a very well written and comprehensive manuscript that gives an excellent overview of the management of patients with multiple myeloma

Minor comments

1) Lines 249-250: ''VEN was also studies'' should be changed to ''VEN was also studied...''

2) Lines 120-122: Authors should present data of ARROW trial in a more clear way

Author Response

Answers to the Reviewer's Comments

It is a very well written and comprehensive manuscript that gives an excellent overview of the management of patients with multiple myeloma

Minor comments

1) Lines 249-250: ''VEN was also studies'' should be changed to ''VEN was also studied...''

Answer: Thank you for your comment, we have corrected this.

2) Lines 120-122: Authors should present data of ARROW trial in a more clear way

Answer: Thank you for your comment, we have modified our manuscript accordingly, Line 141-146.

Reviewer 2 Report

In this review, Dima et al. have comprehensively discuss the state of art of anti-myeloma available treatments and current recommendations. 

However, the focus and principal argument of this review is the evidence-based recommendations in current clinical practice; therefore, sections 2 and 3 on current available therapies with related studies and literature are out of the scope and extend too much this review becoming too long and dispersive for the reader. It would be better to summarize current therapies in one short section, as there are several recent reviews discussing on mechanisms of action, clinical benefits, safety, and mechanisms of resistance of these agents (e.g., De Novellis D, et al. Innovative Anti-CD38 and Anti-BCMA Targeted Therapies in Multiple Myeloma: Mechanisms of Action and Resistance. Int J Mol Sci. 2022 Dec 30;24(1):645. doi: 10.3390/ijms24010645. PMID: 36614086; PMCID: PMC9820921.).

Few typos are present (mg/m2, 2 as superscript).

Author Response

[Answers to the Reviewer’s Comments]

In this review, Dima et al. have comprehensively discuss the state of art of anti-myeloma available treatments and current recommendations. 

However, the focus and principal argument of this review is the evidence-based recommendations in current clinical practice; therefore, sections 2 and 3 on current available therapies with related studies and literature are out of the scope and extend too much this review becoming too long and dispersive for the reader. It would be better to summarize current therapies in one short section, as there are several recent reviews discussing on mechanisms of action, clinical benefits, safety, and mechanisms of resistance of these agents (e.g., De Novellis D, et al. Innovative Anti-CD38 and Anti-BCMA Targeted Therapies in Multiple Myeloma: Mechanisms of Action and Resistance. Int J Mol Sci. 2022 Dec 30;24(1):645. doi: 10.3390/ijms24010645. PMID: 36614086; PMCID: PMC9820921.).

Answer: Thank you very much for your comment, we have made our best effort to reduce the size of the manuscript, and remove sections specifically referring to mechanisms of action, safety and mechanisms of resistance of plasma-cell directed agents.

Few typos are present (mg/m2, 2 as superscript).

Answer: Thank you for your v=comment, we have corrected the typos.

Reviewer 3 Report

The interesting manuscript of Dima et al. detailed the therapeutic modalities available for patients with relapsed-refractory MM by providing a useful overview of the critical factors guiding the selection of appropriate combinations of drugs or cell therapies, in the context of a large and varied therapeutic scenario.

The manuscript is well written, comprehensively referenced, and is suitable for publication.

Author Response

[Answers to the Reviewer' Comments]

The interesting manuscript of Dima et al. detailed the therapeutic modalities available for patients with relapsed-refractory MM by providing a useful overview of the critical factors guiding the selection of appropriate combinations of drugs or cell therapies, in the context of a large and varied therapeutic scenario.

The manuscript is well written, comprehensively referenced, and is suitable for publication

Answer: Thank you very much for your kind words.